

# LLM-powered threat intelligence: a retrieval-augmented generation approach for cyber attack investigation

Abeer Alhuzali

Department of Computer Science, Faculty of Computing and Information Technology,
King Abdulaziz University, Jeddah, Saudi Arabia

## ABSTRACT

Threat intelligence involves collecting, analyzing, and disseminating information about cyber threats to help organizations proactively defend against attacks. However, manually investigating cyberattacks using Cyber Threat Intelligence (CTI) data is challenging due to its heterogeneity, complexity, and volume. While Large Language Models (LLMs) offer potential for automating attack investigation, they suffer from hallucinations, outdated knowledge, and technical misinterpretations. To address these limitations, we propose a Retrieval-Augmented Generation (RAG)-based LLM system called RAGIntel, which enhances accuracy by retrieving and leveraging structured threat intelligence from MITRE ATT&CK. Our approach employs a hybrid retrieval algorithm with reranking and compression strategies to provide precise, context-aware responses. We evaluated RAGIntel on 339 attack investigation queries drawn from diverse benchmarks, using multiple evaluation metrics, and found that it delivers performance comparable to that of standalone LLMs. This study advances automated attack investigation by leveraging RAG-based LLMs, providing a scalable, accurate, and up-to-date solution for cybersecurity analysts.

# INTRODUCTION

Threat intelligence is a systematic process of collecting, evaluating, and sharing actionable information about current, emerging, or potential cyber threats that can impact an organization's assets, operations, or reputation. The intelligence information could be specific about planned attacks or campaigns, such as IoCs and attack vectors, or detailed technical data about tools and techniques used in the attacks, including malware signatures and phishing email characteristics. This intelligence provides actionable insights into adversarial Tactics, Techniques, and Procedures (TTPs), enabling organizations to anticipate and mitigate attacks before they cause harm. It is a proactive approach that transforms raw data into meaningful information, empowering decision-makers to protect their systems effectively. Cyber Threat Intelligence (CTI) data comes from diverse sources such as Open-Source Intelligence (OSINT), intelligence collected from publicly available resources (*e.g.*, industry blogs, security advisories, and social media platforms), the MITRE

Corresponding author
Abeer Alhuzali,
aalhathle@kau.edu.sa

ATT&CK knowledge base, and Common Vulnerabilities and Exposures (CVE), a publicly accessible reference for known security vulnerabilities, *etc.*

Attack investigation involves analyzing these CTI data and inferring several key points, such as the attacker's tools, tactics, techniques, and procedures (TTPs). In addition, identifying the individual or organization behind the attack, known as attack attribution, is a crucial and challenging aspect in the attack investigation process. Gathering, synthesizing, and analyzing the CTI data for attack investigation is not trivial due to data heterogeneity, complexity, size, and time-dependability. Manual procedures are ineffective and time-consuming. Therefore, several automated approaches are proposed in the literature to tackle this problem, such as *Gao et al. (2023a)*, *Liao et al. (2016)*, and *Arikkat et al. (2024)*. With recent advancements in Artificial Intelligence (AI) and particularly Large Language Models (LLMs), innovative solutions for knowledge extraction and reasoning problems, such as attack investigation using CTI data, can be achieved (*Divakaran & Peddinti, 2024*; *Clairoux-Trepanier et al., 2024*). LLMs can employ CTI data and enhance the attack investigation by improving the processing, analysis, and reasoning of massive volumes of unstructured threat data. This enables security analysts to leverage more CTI sources and perform attack investigations effectively. However, LLMs are susceptible to generating hallucinations (*Martino, Iannelli & Truong, 2023*; *Perković, Drobnjak & Botički, 2024*), irrelevant and made-up content to the input data. In addition, LLMs might produce outdated data as LLMs are trained during a particular time, while cyberattacks are consistently evolving. Retraining LLMs more frequently is a complex task. Lastly, LLMs could misinterpret technical texts, resulting in inaccurate or untrustworthy outputs (*Liu et al., 2023*; *Kim et al., 2024*). Given these limitations, using LLMs for attack investigation requires caution, as their tendency to produce false or unreliable intelligence could have severe consequences if applied to real-world cyberattack investigations.

A Retrieval-Augmented Generation (RAG) approach can be used in attack investigation and similar domains to tackle these issues and enhance LLMs' accuracy by incorporating external knowledge bases such as internal organizational data or specialized datasets. For any given user query, RAG-based LLM retrieves the most similar contexts to the user query from the knowledge base. Then, the query and the relevant contexts are supplied to the LLM to generate a response. The system can construct more accurate domain-specific content without further training by integrating relevant information into the generation process. RAG systems offer key benefits. (1) They can improve accuracy and reduce hallucinations by retrieving relevant, up-to-date information from trusted external sources (*e.g.*, threat databases, security reports). (2) Unlike static LLMs, RAG can fetch the latest threat intelligence content from its knowledge base, ensuring responses reflect recent cyber threats, vulnerabilities, and TTPs. This eliminates the need for constant model retraining by leveraging real-time or frequently updated threat data sources. (3) Enhance contextual understanding due to RAG's architecture, combining generative Artificial Intelligence (AI) with retrieval-based search, allowing the model to access domain-specific knowledge (*e.g.*, malware signatures, attack patterns) beyond its pre-trained data. This is particularly

useful in technical domains like cyber attack investigation, where precision and correctness are critical. RAG systems are a powerful solution; however, their effectiveness depends on a high-quality retrieval strategy, knowledge sources, and proper implementation.

Several studies utilized RAGs in the cybersecurity domain, such as *Rajapaksha, Rani & Karafili (2024)*, *Kurniawan, Kiesling & Ekelhart (2024)*, and *Arikkat et al. (2024)*. However, these studies either build a naive RAG system (*Rajapaksha, Rani & Karafili, 2024*) or do not perform a thorough evaluation on large datasets (*Kurniawan, Kiesling & Ekelhart, 2024*). We propose a RAG-based LLM system called RAGIntel to fill this gap. Our tool builds an extensive knowledge database from MITRE ATT&CK, a public knowledge base of known adversary tools, software, campaigns, tactics, techniques, and attack mitigation based on real-world observations. Using a hybrid retrieval algorithm, our tool searches the knowledge base to fetch relevant contexts. The retrieved documents are reranked using a ranking algorithm, compressed using a compression strategy, and finally, the most similar documents for a given query are returned. The generator component of our tool augments the LLM's prompt with the retrieved documents to generate accurate, context-grounded responses. Our tool is evaluated extensively on 339 attack investigation-related queries from various benchmarks (*Alam et al., 2024*) using RAGAS (*Es et al., 2024*). In addition, we compare RAGIntel's performance to standalone LLMs. This article has the following contributions:

- An advanced RAG-based LLM approach to investigate cyber attacks using publicly available cyber threat intelligence data.
- A hybrid retrieval algorithm that employs dense and sparse retrievals. The retrieved contexts are then improved using post-retrieval strategies, namely reranking and compression, to produce the most similar and relevant context to the attack investigation queries.
- An implementation of the approach in a tool called RAGIntel that is publicly available at https://github.com/AbeerAlhuthali/RAGIntel.
- Comprehensive evaluation of the proposed tool using several matrices. In addition, a detailed comparison with standalone LLMs is provided.

The remainder of this article is organized as follows: 'Related Work' provides an overview of related work. 'Research Methodology', 'Implementation Details', and 'Results' describe the methodology employed, implementation details, and discuss the experimental results. Finally, 'Conclusion and Future Work' concludes the study and highlights future research directions.

## RELATED WORK

This section reviews key studies on analyzing cyber threat intelligence (CTI) using retrieval-augmented generation (RAG) systems. We categorize the literature into two broad themes: (1) enhancements to RAG performance and (2) applications of RAG in the cybersecurity domain.

## RAG performance enhancement studies

*Gao et al. (2023b)* and *Fan et al. (2024)* surveyed several research works on building and optimizing RAG systems. For example, *Cuconasu et al. (2024)* studied the influence of the retriever component on the overall performance of RAG systems. In particular, they have conducted experiments to investigate the effects of the quality of the retrieved documents, the number of retrieved contexts, and their position within the prompt. They concluded that including random contexts (noise) in the prompt increases the accuracy of the LLM responses. Additionally, they found that the context that contains the ground truth answer should be placed near the query in the prompt. These authors counterintuitive results suggest that further research is necessary in this area to gain a deeper understanding of RAG behavior. A similar study by *Liu et al. (2024)* investigated how the position of the retrieved contexts affects the performance of LLMs. The study concludes that the performance of LLMs decreases significantly when the relevant content is placed in the middle of the retrieved documents. They have also found that retrieving more context is not always better. In our work, we build upon this approach and retrieve the top 10 most similar documents. We then apply a post-retrieval strategy to further reduce the number of relevant documents to three. *Zhao et al. (2024)* investigated the factors that affect the performance of RAG systems. They performed several experiments utilizing three Question-Answering (QA) datasets and two LLMs, with the aim of understanding the impact of the following factors on RAG systems: retrieval document type, retriever recall, document (context) selection, and prompt engineering techniques. Their study provides suggestions for improving the performance of RAG systems. Our scope differs from that of the abovementioned studies, which focused on understanding the factors that contribute to improving the performance of RAG applications. Nonetheless, we have utilized some of these findings.

The retriever in RAGs is a critical component, and its performance directly affects the generated outcomes. In the following, we discuss key works that studied the role of the retriever in RAG applications. RETRO (*Borgeaud et al., 2022*) is a system that uses a frozen LLM with an external retrieval mechanism for each generation step. It scales to trillions of tokens in the retrieval *corpus*. RETRO demonstrated high efficiency and performance without increasing LLM size. REPLUG (*Shi et al., 2023*) is a framework that allows plug-and-play retrieval augmentation for pretrained LLMs. This approach enhances retrieval quality without requiring retraining of the generator or retriever. It made RAG adoption more flexible in production settings. Several pre- and post-retrieval methods have been introduced to optimize retrieval performance by enhancing both the input queries and the algorithm's output (*Tan et al., 2024*; *Ma et al., 2023*; *Zhuang et al., 2023*). Query rewrite (*Ma et al., 2023*) is a system that focuses on improving the quality of a query by asking the LLM to rewrite the query for the retriever. Their approach outperforms naive RAG or generation models. Query2doc, proposed by *Wang, Yang & Wei (2023)*, is another pre-retrieval strategy. Query2doc presents a query expansion approach that uses LLMs and then generates pseudo-documents using few-shot prompts. These generated documents are integrated with the queries and used by sparse or dense retrievers. Experimental results

show that this approach is practical and can boost the performance of the retrievals. Their approach outperforms naive RAG and generation models. Several articles have discussed how the retrieval outcomes can be improved through post-retrieval strategies, including re-ranking and context compression. Re-ranking the retrieved context aims to re-evaluate the already retrieved contexts and reorder them based on their similarity to the problem. *Zhuang et al. (2023)* introduced a ranking system that utilizes LLM-based Query Likelihood Models (QLMs) and a hybrid zero-shot retriever, demonstrating effectiveness on LLM responses. *Xu, Shi & Choi (2024)* introduced Retrieve, Compress, Prepend (RECOMP), which compresses the retrieved contexts by summarizing them and feeds the summaries to the LLM along with the query. *Hofstätter et al. (2023)* proposed an approach that compresses the encoded vectors per retrieved context before aggregating them and feeding them to the decoder based on a lighter version of Fusion-in-Decoder (FiD). Their approach includes a re-ranking strategy that ranks the retrieved results before applying them to the generator component. Another line of research has explored the concept and impact of making RAG systems modular. *Gao et al. (2024)* introduced a modular RAG framework to tackle the limitations of traditional linear RAG systems. They decomposed a RAG system into independent modules for continued evolution and practical deployment. *Shi et al. (2024)* developed another modular RAG system to enhance the overall response accuracy. Our tool adopts retrieval enhancement strategies, such as context reranking and compression.

Other studies have examined how dataset selection influences the evaluation of RAG applications. For instance, CRAG (*Yang et al., 2024*) is a comprehensive benchmark designed to evaluate RAG systems more effectively, than current benchmarks, which do not fully capture real-world Question Answering (QA) tasks. *De Lima et al. (2024)* studied the role of the datasets in RAG evaluation. They constructed synthetic datasets and proposed strategies for generating such datasets based on the type of user interactions (*e.g.*, reasoning, summarization) for evaluating RAG performance. Here, we selected datasets explicitly designed for attack investigation.

## RAG applications in cybersecurity

*Rajapaksha, Rani & Karafili (2024)* proposed a QA model designed to assist cybersecurity analysts in investigating and attributing cyberattacks by leveraging RAGs and LLMs. Their proposed approach outperforms stand-alone GPT-3.5 and GPT-4o by reducing hallucinations and providing verifiable references. *Yamin et al. (2024)* explored using LLMs and RAG to generate realistic cybersecurity exercise scenarios. *Simoni et al. (2025)* developed the Morse framework, which uses two variations of the RAG architecture to provide answers for cybersecurity questions. The framework incorporates parallel retrieval algorithms to accelerate the retrieval process. CyKG-RAG (*Kurniawan, Kiesling & Ekelhart, 2024*) is a RAG framework that utilizes Knowledge Graphs (KGs) built from cybersecurity knowledge bases, such as MITRE, for cyber threat detection. The KGs enhance the retriever, which captures the semantic relationships within the KGs. A similar approach proposed by *Jeon, Koo & Kim (2024)* that integrates graph models with RAG for cyber threat tracing and investigation. *Munir et al. (2024)* explored using a multimodal

RAG system to effectively predict cyberthreats in transportation systems. In a similar vein, the RAG-based chatbot IntellBot (*Arikkat et al., 2024*), was designed for answering cybersecurity questions, using a knowledge base built from diverse sources. Compared to these systems, our work distinguishes itself in three key aspects: (1) We conduct a comprehensive evaluation using 110 diverse queries, compared to IntellBot's 5-query assessment), (2) we incorporate hybrid retrieval strategies combining both semantic and keyword-based search, and (3) we implement post-retrieval refinement techniques to enhance answer quality. These methodological advancements enable a more rigorous and scalable evaluation of RAG performance in cybersecurity applications.

## THREAT MODEL

Our system investigates cyberattacks documented in curated CTI sources, focusing on extracting attack techniques and attributing threat actors from descriptive text. The *MITRE ATT&CK* repository serves as the source of our knowledge base, covering a broad spectrum of real-world threats, including malware campaigns, phishing, ransomware, supply-chain compromises, cloud intrusions, and ICS-targeted attacks. Adversaries may aim to exfiltrate data, disrupt services, gain persistence, or compromise systems. We assume that the CTI records are accurate and complete, reflecting peer-reviewed threat intelligence. Our system does not simulate live attacks; instead, all scenarios are grounded in documented incidents.

The *MITRE ATT&CK* repository, which curated and peer-reviewed content, significantly reduces the risk of incorporating maliciously crafted or misleading information compared to unverified sources. Nonetheless, recent studies (*Zhang et al., 2025*) have shown that RAG systems may be susceptible to knowledge poisoning attacks, in which adversaries inject false or misleading information into the underlying knowledge base to influence outputs. Although our reliance on *MITRE ATT&CK* mitigates this threat to some extent, it does not eliminate it. Future work will explore additional safeguards, such as content verification pipelines, to further strengthen resilience against such attacks.

## RESEARCH METHODOLOGY

### System workflow

Our approach leverages the strengths of LLMs while mitigating some of their limitations, such as hallucinations, through a RAG framework. As illustrated in Fig. 1, our RAG-based cyber threat intelligence system operates in two key phases:

- Retriever: Searches external knowledge bases to fetch relevant context using a hybrid retrieval algorithm (steps 1–5 in Fig. 1), ranking, compressing, and returning the most similar documents for a given query.
- Generator: Augments the LLM's prompt with the retrieved documents to generate accurate, context-grounded responses (shown in the "Generator" section of the workflow).

By integrating the retriever and the generator, the system ensures responses are both informed by authoritative sources and refined by the LLM's reasoning capabilities.

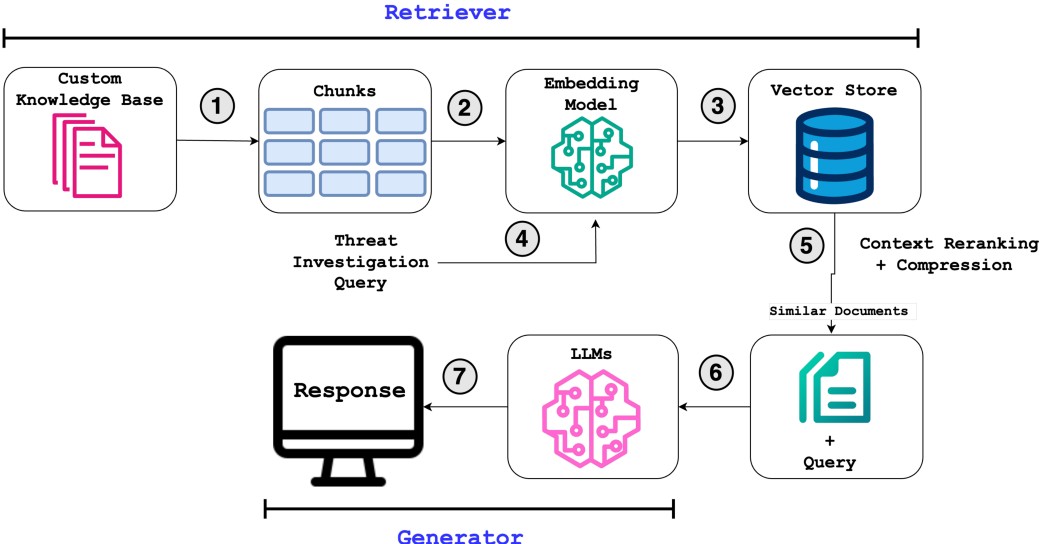

**Figure 1 Overall workflow of our approach implemented in RAGIntel.**

## Retriever component

### Knowledge base

One essential step in any RAG-based system is building a knowledge base related to the problem being investigated. In RAGIntel, we utilize the *MITRE ATT&CK* (*MITRE Corporation, 2025*), a publicly accessible knowledge base of adversary behavior. It documents common tactics, techniques, and procedures attackers use for various platforms such as enterprises, mobile, cloud, Windows, macOS, and industrial control systems (ICS). The content included in *MITRE ATT&CK* highlights emerging threats and guides the development of effective defensive strategies. In our implementation, the knowledge base includes detailed data about assets, attack techniques, tactics, mitigations, known attack campaigns, software used in attacks, and attack groups. While the knowledge base provides rich and extensive coverage of relevant CTI information, it is not intended to encompass all possible categories or sources of CTI data.

All assets included in our knowledge base are related to the industrial control systems domain (14 assets). Each asset is annotated with its name, description, and unique ID. RAGIntel's knowledge base contains a thorough set of attack techniques representing how an adversary achieves a goal by performing an action. For example, the attacker may perform a Denial of Service (DoS) attack to block the availability of a service. In RAGIntel, there are a total of 203 techniques and 453 sub-techniques related to enterprises, 73 techniques and 46 sub-techniques pertaining to mobile, and 83 techniques related to attacking ICSs are included. Similarly, we incorporate information about various attack tactics used to attack enterprises, mobile systems, and ICSs extracted from *MITRE ATT&CK*. Tactics convey the attacker's goal. Our knowledge base has 14 enterprise, 14 mobile, and 12 ICS tactics. In addition, many mitigation techniques that are used to prevent the attack techniques from being successful are included in RAGIntel's knowledge

| Example of | ID | Name | Discription |
|---|---|---|---|
| **asset** | A0014 | Routers | A computer that is a gateway between two networks at OSI layer 3 and that relays and directs data packets through that inter-network. The most common form of router operates on IP packets. |
| **Enterprise Technique** | T1136 | Create Account | Adversaries may create an account to maintain access to victim systems. With a sufficient level of access, creating such accounts may be used to establish secondary credentialed access that do not require persistent remote access tools to be deployed on the system. |
| **Sub-technique** | T1136.001 | Local Account | Adversaries may create a local account to maintain access to victim systems. Local accounts are those configured by an organization for use by users, remote support, services, or for administration on a single system or service. |
| **Mobile tactic** | TA0027 | Initial Access | The adversary is trying to get into your device. |
| **ICS mitigation** | M0948 | Application Isolation and Sandboxing | Restrict the execution of code to a virtual environment on or in-transit to an endpoint system. |
| **Attack campaign** | C0025 | 2016 Ukraine Electric Power Attack | 2016 Ukraine Electric Power Attack was a Sandworm Team campaign during which they used Industroyer malware to target and disrupt distribution substations within the Ukrainian power grid. This campaign was the second major public attack conducted against Ukraine by Sandworm Team. |

| | ID | Name | Associated Groups | Description |
|---|---|---|---|---|
| **Example of attack group** | G0008 | Carbanak | Anunak | Carbanak is a cybercriminal group that has used Carbanak malware to target financial institutions since at least 2013. Carbanak may be linked to groups tracked separately as Cobalt Group and FIN7 that have also used Carbanak malware. |

| | ID | Name | Associated Software | Description |
|---|---|---|---|---|
| **Example of attack software** | S0504 | Anchor | Anchor_DNS | Anchor is one of a family of backdoor malware that has been used in conjunction with TrickBot on selected high profile targets since at least 2018. |

**Figure 2 Examples of CTI types included in RAGIntel's knowledge base.**

base. Specifically, 44 enterprise mitigations, 13 mobile, and 52 ICS mitigations are included. Furthermore, having information about attack groups is essential for conducting thorough attack investigations. Therefore, we retain detailed information about known attack groups, including a detailed description of 163 attack groups, in our knowledge base. As shown in Fig. 2, each group has a name and possibly other known aliases of the same group collected from public reports. We also have various known attack campaigns (36 instances) and software (826 instances) in our knowledge base. The list of software represents known software names, IDs, and descriptions collected from public sources. Some software has multiple names associated with the same instance. Lastly, the MITRE ATT&CK uses the term campaign to describe a set of coordinated cyber intrusions occurring within a particular period, directed at common targets and having common objectives. Figure 2 illustrates representative examples of CTI types leveraged in our system; it is not intended to be an exhaustive enumeration of all possible CTI categories.

Each of the seven CTI types (*i.e.*, assets, attack techniques, tactics, mitigations, software, groups, and campaigns) is fed into RAGIntel as an HTML file extracted from *MITRE ATT&CK* repository.

### Chunking and the embedding model

Before utilizing the embedding model, RAGIntel breaks down the files in the knowledge base into chunks. Chunking files enables their use with embedding models, which have specific input sizes. Moreover, without chunking, each file will contain only one embedding, which fails to provide relevant context. The chunking strategy employed in our work is hierarchical recursive splitting, which involves two levels of chunking to capture different levels of contexts: parent and child splitting. The parent splitting creates larger chunks of 2,000 characters to capture the broad context. Then, each chunk is further split into 500-character chunks for increased granularity. The parent-child relationships are maintained using metadata. Next, we generate embeddings from the text chunks, converting them into numerical representations (vectors) and storing these in a vector database. The embedding model maps each chunk into a multidimensional space, arranging data points based on semantic similarity; thus, closely related content is positioned closer to one another. Selecting the embedding model is crucial in RAG systems, as their performance depends heavily on the quality of the context retrieved from the vector database.

### Vector store and hybrid retrieval algorithm

The embeddings generated in the previous step are transformed into vectors and stored in the vector database, which serves as a repository for answering the user's queries. The vector database stores the generated vector embeddings, the original chunks, and their associated metadata. The vector store can be updated later if additional information becomes available. The process of vectorizing the data enables semantic vector search to find data points in the store similar to an input. Upon receiving a query, the system converts it into embeddings and searches the vector store for similar embeddings. Therefore, the quality of the retrieval process affects the performance of RAGIntel in general. We employ a hybrid retrieval approach that combines *BM25* (*Robertson & Zaragoza, 2009*) and *dense* embeddings. BM25 is a typical keyword-based retrieval algorithm for sparse retrieval, while dense retrieval employs approximate nearest neighbors. Results from both are merged by the hybrid retrieval algorithm, duplicates are removed *via* content hashing, and the final list is returned. In addition, this list of retrieved documents (*i.e.*, 10 documents) is further reranked using Langchain's *FlashrankRerank*. Finally, we apply a compression strategy using Langchain's *ContextualCompressionRetriever* to compress the number of retrieved documents to the top 3 most similar documents, which is considered the query's final *retrieved context*.

## Generator component

The retrieved knowledge from the retrieval component is integrated with the pre-trained LLMs to augment their contextual understanding. This synthesis enables the generation of

higher-quality responses with improved accuracy, depth, and engagement. The following sections present the LLMs and prompt components.

### RAG queries and prompts

Prompts are used to interact and communicate with LLMs. Therefore, queries are automatically sent to our RAG-based system through prompts. We adopted two datasets from CTIBench (*Alam et al., 2024*), a public benchmark of datasets for evaluating LLMs in the cyber threat intelligence domain, specifically CTI-ATE and CTI-TAA. The CTI-ATE contains 60 queries while the other includes 50 prompts. Additionally, we utilize a third dataset, CTI-ATTACK (*dattaraj, 2025*), which consists of 299 attack-related questions.

Generally, each of these queries and prompts represents an attack investigation question that a cyber analyst must answer to investigate a cyberattack or evaluate a cyber threat. Each of the prompts is automatically sent to the LLMs in addition to the most similar context to the query retrieved from the vector store by our hybrid retrieval algorithm. The LLM then responds to the provided query, taking into account the retrieved context. The response is automatically recorded and outputted. Note that all prompts used in this work are zero-shot prompts, meaning that responses are generated without any prior task-specific examples or fine-tuning, relying solely on the LLM's pre-trained knowledge, the provided instructions, and retrieved context. The nature of the queries, examples, and details of the datasets are provided in 'Evaluation Datasets'.

### Large language models (LLMs)

In our work, we employ various pre-trained LLMs to generate query responses. Two inputs are needed for the LLM to work in RAGIntel: a threat intelligence investigation prompt and the most similar context to the query retrieved from the vector store. Given this input, each LLM will generate a response. The LLMs used in RAGIntel are OpenAI GPT-4o mini (*OpenAI, 2025a*) and GPT-3.5 turbo (*OpenAI, 2025b*), Meta BART (*Lewis et al., 2020*), and Google Flan-T5 (*Chung et al., 2022*; *Sanh et al., 2019*). We assess our RAG system using more than one LLM to mitigate model-specific biases, leverage complementary strengths (*e.g.*, BART's comprehension, Flan-T5's instruction-following, GPT-4o's reasoning), and ensure robustness across different architectures.

## Evaluation datasets

We utilized three datasets focused on cyber attack investigation. Two are from the CTIBench (*Alam et al., 2024*, *2025*) benchmark (*i.e.*, CTI-ATE and CTI-TAA), and the third one is CTI-ATTACK (*dattaraj, 2025*). The CIT-ATE dataset contains 60 attack descriptions collected from public sources. The dataset includes a prompt for each sample text that asks LLM to identify attack techniques according to *MITRE ATT&CK*. It also has ground truth answers and other labels. Figure 3 illustrates an example from the original CTI-ATE dataset showing only the prompt and the corresponding ground truth. To utilize this dataset for our work, we retained only the essential labels (*i.e.*, text, prompt, and ground truth). We then removed the technique IDs from all prompts, as they are already included in our knowledge base. Most importantly, we aim to evaluate our RAG retrieval

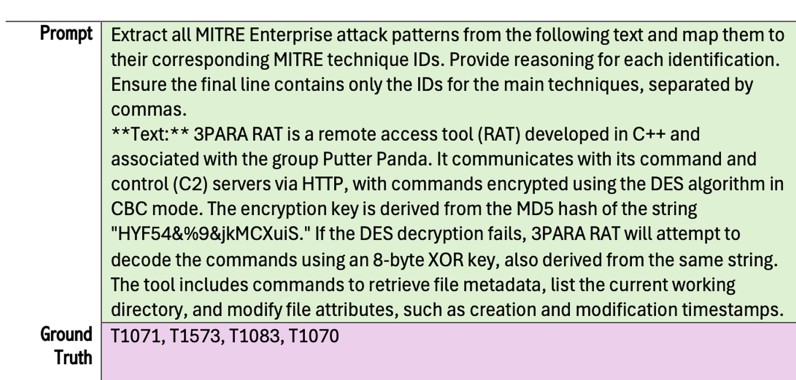

**Prompt** | Extract all MITRE Enterprise attack patterns from the following text and map them to their corresponding MITRE technique IDs. Provide reasoning for each identification. Ensure the final line contains only the IDs for the main techniques, separated by commas, excluding any subtechnique IDs. MITRE Enterprise IDs are given below as reference.
**Text:** 3PARA RAT is a remote access tool (RAT) developed in C++ and associated with the group Putter Panda. It communicates with its command and control (C2) servers via HTTP, with commands encrypted using the DES algorithm in CBC mode. The encryption key is derived from the MD5 hash of the string "HYF54&%9&jkMCXuiS." If the DES decryption fails, 3PARA RAT will attempt to decode the commands using an 8-byte XOR key, also derived from the same string. The tool includes commands to retrieve file metadata, list the current working directory, and modify file attributes, such as creation and modification timestamps.
**List of All MITRE Enterprise technique IDs** ID : Name T1548 : Abuse Elevation Control Mechanism T1134 : Access Token Manipulation T1531 : Account Access Removal T1087 : Account Discovery T1098 : Account Manipulation (List truncated for space)

**Ground Truth** | T1071, T1573, T1083, T1070

**Figure 3 Example taken from CTI-ATE dataset (without modifications).**

**Prompt** | Extract all MITRE Enterprise attack patterns from the following text and map them to their corresponding MITRE technique IDs. Provide reasoning for each identification. Ensure the final line contains only the IDs for the main techniques, separated by commas.
**Text:** 3PARA RAT is a remote access tool (RAT) developed in C++ and associated with the group Putter Panda. It communicates with its command and control (C2) servers via HTTP, with commands encrypted using the DES algorithm in CBC mode. The encryption key is derived from the MD5 hash of the string "HYF54&%9&jkMCXuiS." If the DES decryption fails, 3PARA RAT will attempt to decode the commands using an 8-byte XOR key, also derived from the same string. The tool includes commands to retrieve file metadata, list the current working directory, and modify file attributes, such as creation and modification timestamps.

**Ground Truth** | T1071, T1573, T1083, T1070

**Figure 4 Sample of the modified dataset used in RAGIntel, corresponding to the example in Fig. 3, with the highlighted sections removed.**

component's ability to retrieve the relevant contexts from the vector store. Figure 4 demonstrates these updates.

In addition, CTI-TAA contains 50 entries, each representing a publicly available threat report labeled as `text`, source of the report labeled as `URL`, and `prompt`, which asks LLM to identify the attributing threat actors or malware families from each attack described in the `text`. The ground truth answer for each query is crucial for evaluating any RAG-based system. Since the original dataset did not include this label, we added the ground truth answers after reviewing each original attack description from its corresponding `URL`. Figure 5 illustrates a sample of this updated dataset, showing a prompt and the corresponding ground truth.

CTI-ATTACK dataset originally contains 281 short attack descriptions along with their ground truth answers, which are the MITRE technique IDs. The dataset has no prompts (unlike CTIBench), no description of how the data was collected and verified, and it is in

| Prompt | You are given a threat report that describes a cyber incident. Any direct mentions of threat actor groups, campaign names, or malware names have been replaced with [PLACEHOLDER].<br>Your task is to attribute the incident to a known threat actor or malware. Map it to relevant MITRE ATT&CK threat groups (e.g., APT36) or Software/Tools (e.g., Crimson RAT - S0334)<br>Threat Report: [PLACEHOLDER], a notorious name in the realm of cyber threats, has loomed large over the digital landscape since its inception in 2014. Originally identified as a banking Trojan focused on financial data theft, [PLACEHOLDER] has evolved into a highly adaptable and multifaceted malware, capable of causing widespread disruption to both individuals and organizations alike.   In this comprehensive analysis, we embark on a journey into the intricate workings of [PLACEHOLDER], meticulously dissecting its tactics, functionalities, and the imminent dangers it presents.  (truncated for space) |
| URL | https://medium.com/@zyadlzyatsoc/comprehensive-analysis-of-emotet-malware-part-1-by-zyad-elzyat-35d5cf33a3c0 |
| Ground Truth | Emotet |

**Figure 5  Example from the updated CTI-TAA dataset.**

| Prompt | Extract all MITRE attack patterns from the following text and map them to their corresponding MITRE technique IDs. Provide reasoning for each identification. Ensure the final line contains only the IDs for the main techniques, separated by commas, excluding any subtechnique IDs.<br>**Text:** Detected a suspicious PowerShell script execution on one of our workstations. The script seems to be collecting system information and attempting to establish an outbound connection. |
| Ground Truth | T1059.001, T1082, T1571 |

**Figure 6  Example from the updated CTI-ATTACK dataset.**

JSON format. Therefore, we perform a preprocessing step in which we convert it to a CSV format, add a prompt to each attack description, and remove duplicate entries. The final dataset after the preprocessing has 229 attack descriptions. Figure 6 illustrates an example from the updated CTI-ATTACK dataset.

In summary, these updated datasets will help in assessing the effectiveness of our RAG-based system in performing threat investigations, as we will explain in 'Results'.

## IMPLEMENTATION DETAILS

RAGIntel was built using Python with 400 total lines of code, and we have utilized several libraries. Specifically, HTML files in our Knowledge base are fetched from the MITRE ATT&CK website and stored locally. RAGIntel automatically retrieves these files and breaks them down into chunks using *Langchain's RecursiveCharacterTextSplitter* function. Our parent chunk size is set to 2,000, and an overlap of 200. The child split size is set to 500, and an overlap of 50 characters. Next, the embedding model creates the embedding vectors from the chunks. For the embedding model, we utilized the *nomic-ai/nomic-embed-text-v1.5* model from the Hugging Face (*Face, 2025*). Then, the resulting vectors are stored in the vector store. We used *FAISS* (*Faiss, 2025*), a library that

enables storing and searching similar documents, as a vector store. LangChain's BM25 (*LangChain, 2025a*) retriever was utilized in our hybrid retrieval algorithm. The reranker is Langchain's *FlashrankRerank* (*LangChain, 2025c*), and the compression strategy is done through Langchain's *ContextualCompressionRetriever* (*LangChain, 2025b*) to compress the number of retrieved documents. We incorporated four different LLMs: OpenAI GPT-4o mini and GPT-3.5 turbo, Meta BART base model, and Google Flan-T5 small model. All prompts used in this work are zero-shot prompts in which LLMs are instructed to act as cybersecurity experts. For evaluating RAGIntel, we used a well-known evaluation metric called RAGAS version 0.2.13 (*Es et al., 2024*). In 'Results', we provide detailed information related to RAGAS.

# RESULTS

## Experimental setup

We implemented and evaluated our system using Google Colab (*Google, 2017*) connected to Python 3 Google Compute Engine backend (TPU) and a total of 334.56 GB of memory. Each of the updated datasets is processed and evaluated individually. We assessed each dataset using the four LLMs: OpenAI GPT-4o mini and GPT-3.5 turbo, BART base model, and Flan-T5 small model. The temperature for all the models was set to zero for deterministic responses. The experiments were conducted and reported over a 5-day period from April 22, 2025.

## Evaluation metrics

To evaluate all tasks, we use RAGAS, a library for evaluating LLMs' applications. RAGAS requires four main pieces of information: the prompt, the LLM's response to the prompt, the retrieved contexts, and the ground truth answer to the prompt. Our updated datasets include a list of prompts and their corresponding ground truth answers. The retriever component of RAGIntel retrieves the relevant contexts for each prompt. The LLMs' responses are acquired and recorded for each prompt and the three most similar contexts, as illustrated in Fig. 1. RAGAS provides a set of evaluation metrics specifically designed to measure the performance of RAG-based systems. These metrics aim to measure the performance of various aspects of RAG applications objectively. We have used six metrics to measure the retriever and generator components. Specifically, these include context precision, context recall, noise sensitivity, response relevancy, factual correctness, and faithfulness. Below, we provide a brief explanation of each.

**Evaluating the retriever component**. RAGAS provides several metrics to assess the performance of the retriever. We have evaluated RAGIntel's retriever using:

- **Context precision** conveys the quality of the retrieval pipeline. Therefore, it is a metric that calculates the relevance of the parts in the retrieved contexts. It is derived from rewarding systems that rank relevant chunks higher by averaging precision at each step, weighted by their relevance. It is calculated using the query, ground truth, and contexts, with values ranging from 0 to 1.

- **Context recall** calculates how much relevant information is successfully retrieved in response to a query compared to a ground truth. This score ranges from 0 to 1, where a higher score indicates fewer relevant contexts were not retrieved. In our implementation, we used LLM-based context recall measured using the following equation:

$$\text{Context Recall} = \frac{\text{Number of claims in the ground truth supported by the retrieved context}}{\text{Total number of claims in the ground truth}}. \quad (1)$$

- **Noise sensitivity** of relevant context quantifies how often a system produces wrong answers when working with relevant contexts. Score ranges from 0 (best) to 1 (worst). To compute noise sensitivity, RAGAS verifies each response generated by an LLM for two key aspects: (a) factual accuracy against the ground truth, and (b) drivability from the provided context. An ideal response contains only claims satisfying both conditions.

$$\text{Noise Sensitivity} = \frac{\text{Total number of incorrect claims in response}}{\text{Total number of claims in the response}}. \quad (2)$$

**Evaluating the generator component**. RAGAS provides a set of metrics designed to assess the quality of the generator. We incorporated several of them as explained below.

- **Faithfulness** metric assesses how relevant a generated response by LLM is to the retrieved context. This is measured by verifying whether the retrieved context supports the statements made in the responses

$$\text{Faithfulness} = \frac{\text{Number of claims in the response supported by the retrieved context}}{\text{Total number of claims in the response}}. \quad (3)$$

- **Response relevancy** measures how relevant a generated response is to a query, without evaluating the factual accuracy of the response itself. The score increases when the response more accurately reflects the query, while it penalizes responses that are lacking key details or include unnecessary information.

- **Factual correctness** measures the alignment between a generated response and a ground truth, scoring factual accuracy on a normalized scale (0–1). Higher values denote stronger agreement with the ground truth. It can be calculated using the precision, recall, or F1 formulas below. Our calculations of this metric are based on the F1-score.

$$\text{Precision} = \frac{TP}{(TP + FP)} \quad (4)$$

$$\text{Recall} = \frac{TP}{(TP + FN)} \quad (5)$$

$$\text{F1-score} = \frac{2 \times Precision \times Recall}{(Precision + Recall)} \quad (6)$$

where True Positive (TP) denotes the number of claims in response that are present in the ground truth, False Positive (FP) indicates the number of claims in response that are not present in the ground truth, and False Negative (FN) indicates the number of claims in the ground truth that are not present in the response.

**Table 1 Models' performance across datasets and metrics.**

| Dataset | Model | FC | RR | NS | F | CR | CP |
|---|---|---|---|---|---|---|---|
| CTI-ATE | GPT-3.5 | 0.150 | 0.669 | 0.065 | 0.140 | 0.465 | 0.790 |
| | GPT-4o | 0.153 | 0.843 | 0.078 | 0.145 | | |
| | BART | 0.063 | 0.000 | 0.224 | 0.345 | | |
| | Flan T5 | 0.005 | 0.595 | 0.173 | 0.256 | | |
| CTI-TAA | GPT-3.5 | 0.037 | 0.791 | 0.011 | 0.036 | 0.344 | 0.091 |
| | GPT-4o | 0.057 | 0.826 | 0.000 | 0.070 | | |
| | BART | 0.000 | 0.413 | 0.000 | 0.140 | | |
| | Flan T5 | 0.094 | 0.441 | 0.000 | 0.283 | | |
| CTI-ATTACK | GPT-3.5 | 0.115 | 0.713 | 0.029 | 0.118 | 0.418 | 0.340 |
| | GPT-4o | 0.186 | 0.684 | 0.023 | 0.069 | | |
| | BART | 0.023 | 0.481 | 0.035 | 0.169 | | |
| | Flan T5 | 0.017 | 0.074 | 0.027 | 0.100 | | |

**Note:**
FC, Factual correctness; RR, response relevancy; NS, noise sensitivity; F, faithfulness; CR, context recall; CP, context precision.

## Results summary

Table 1 below summarizes the evaluation results of RAGIntel across all datasets. For the CTI-ATE dataset, GPT-4o was the best-performing model across all models in terms of factual correctness and response relevancy. This indicates that it is the best model in generating responses relevant to the ground truths and queries. The average context recall and context precision for all 60 queries are relatively high, at 0.465 and 0.790, respectively. This implies that the retrieval algorithm was effective. Note, context recall and context precision are computed based on the queries and the ground truths; therefore, they are irrelevant to the LLMs' responses. As a result, these metrics are calculated globally to reflect the overall retrieval accuracy across all queries.

On the CTI-TAA dataset, GPT-4o leads in response relevancy and noise sensitivity, while Flan-T5 excels in factual correctness and faithfulness. The average context recall and context precision for all 50 queries are low, at 0.344 and 0.091, respectively. These low scores suggest that the retriever frequently retrieved irrelevant passages, which likely degraded downstream task performance. We provide further discussion on the results in 'CTI-TAA Results'.

The CTI-ATTACK dataset contains relatively concise attack descriptions compared to the CTI-ATE and CTI-TAA datasets. The results of this dataset indicate that GPT-3.5 provides a trade-off among accuracy, relevance, and faithfulness. In contrast, GPT-4o performs well in terms of factual correctness, although this comes at the cost of adhering to the context.

The total runtime of RAGIntel across all datasets was 6 h 0 m 33 s. The majority of this time was consumed by LLM response generation and subsequent RAGAS evaluation. The time required for constructing the knowledge store and initializing the retriever for all datasets was 11 m 40 s. LLM response generation took 1 h 10 m 40 s for CTI-ATE, 50 m 9 s for CTI-TAA, and 1 h 35 m 17 s for CTI-ATTACK. RAGAS evaluation required 45 m 3 s,

37 m 10 s, and 50 m 34 s for CTI-ATE, CTI-TAA, and CTI-ATTACK, respectively. The difference in LLM generation times is partly due to the length and complexity of the attack descriptions in the query sets. The CTI-ATE and CTI-TAA datasets contain significantly longer and more detailed attack descriptions (*i.e.*, more tokens per query) compared to the CTI-ATTACK dataset. This observation underscores that the runtime of RAGIntel is influenced by the number of queries and the verbosity and complexity of their content.

## CTI-ATE results

As shown in Table 1, GPT-4o scores the best among all other models in response relevancy (0.843) and factual correctness (0.153). This indicates that it generates answers closely aligned with the queries themselves (response relevancy) and with the ground truths (factual correctness). Additionally, it handles noise effectively, as its low score indicates that it does not produce incorrect responses when having relevant contexts. Although the factual correctness score is the highest compared to the other models, it is generally considered low, as the value of factual correctness ranges from 0 to 1. Similarly, its faithfulness score is low, implying that the responses deviate from the retrieved contexts. To understand the reasons behind the low scores in factual correctness and faithfulness, we examined the model-generated responses in comparison to both the ground truth answers and the retrieved contexts for each query. Our analysis revealed that the responses were generally of high quality. The low scores stem from two main factors: (1) the ground truth answers are very brief, consisting only of lists of technique IDs without any reasoning (as illustrated in Fig. 4), whereas the LLM-generated responses include both the technique IDs and explanatory reasoning; (2) the relevant techniques are not explicitly mentioned in the queries and must be inferred. The retriever component is designed to select contexts most similar to the query, which RAGIntel successfully achieves.

BART performed well regarding faithfulness (0.345), indicating a better correlation between the ground truths and the retrieved context. However, it struggles severely with response relevancy (0.00), meaning its responses are often irrelevant to the queries. In addition, its factual correctness (0.063) is low, similar to Flan T5 (0.016), indicating that it frequently generates incorrect facts compared to the ground truths. However, FLAN-T5's performance degrades with noisy inputs (*i.e.*, noise sensitivity is 0.224). GPT-3.5 balances all metrics except for faithfulness (0.140).

## CTI-TAA results

Table 1 demonstrates that Flan-T5 and GPT-4o are the best-performing models on the TAA dataset. The noise sensitivity and response relevancy evaluation results were acceptable for all models. To assess the retrieval strategy on this dataset, we examine the context precision, recall, and noise sensitivity. The context precision and recall are low, averaging at 0.344 and 0.091, respectively. The retrieved context is more closely aligned with the ground truth (context recall 0.344) than with the query (precision 0.091). We argue that this is not surprising for the nature of the queries in this dataset, which are different than the ones in CTI-ATE dataset. Queries in the CTI-TAA dataset require implicit inference of attack attribution (*e.g.*, linking attack patterns to groups not explicitly

**Table 2 Evaluation results of the sample query from the CTI-TAA dataset.**

| Metric | GPT-3.5 | GPT-4o | BART | Flan T5 |
|---|---|---|---|---|
| Faithfulness | 0.125 | 0.188 | 0.776 | 0 |
| Factual correctness | 0 | 0 | 0 | 0 |
| Response relevancy | 0.821 | 0.849 | 0.715 | 0.816 |
| Noise sensitivity | 0 | 0 | 0 | 0 |
| Context recall | | 1 | | |
| Context precision | | 0 | | |

named in the context). The knowledge base, from which the contexts are retrieved, has various attack groups and a high-level description of their activities. In addition, attack groups commonly have multiple aliases, and MITRE ATT&CK might not have the name that matches the one in the ground truth (*e.g.*, "Fancy Bear" *vs* "APT28"). In addition, it is common to have different attack groups that have activity overlaps (*e.g.*, ransomware deployment by both APT29 and APT41), which may create retrieval ambiguity. Regarding the generation assessment of our RAG system, the response relevance is very high in the GPT models ($\approx 0.8$) and moderately high in BART and Flan-T5 ($\approx 0.4$).

On the other hand, generally, all models underperform in faithfulness and factual correctness, indicating that models' responses deviate from the contexts and the ground truth. Two factors might cause the low scores. First, there is a possibility of hallucinations. Second, the context retrieved for a query does not have the ground truth answer. It is worth mentioning that a small change between the response and the context or ground truth causes changes in the overall value of these metrics, as explained in Eqs. (3) and (6). The first case is handled by setting the temperature for all models to 0 for deterministic responses.

The following example demonstrates this behavior on a sample query from the dataset and its evaluation results (refer to Table 2).

- **Prompt:** You are given a threat report that describes a cyber incident. Any direct mentions of threat actor groups, campaign names, or malware names have been replaced with [PLACEHOLDER]. Your task is to attribute the incident to a known threat actor or malware. Map it to relevant MITRE ATT&CK threat groups (*e.g.*, APT36) or Software/Tools (*e.g.*, Crimson RAT—S0334).
  Threat Report: "Prolific Iranian advanced persistent threat group (APT) [PLACEHOLDER] has repeatedly targeted several Israeli organizations throughout 2022 in cyberattacks that were notable for leveraging a series of custom downloaders that use legitimate Microsoft cloud services to conduct attacker communications and exfiltrate data. [PLACEHOLDER] in the attacks deployed four specific new downloaders, SampleCheck5000 (SC5k v1-v3), ODAgent, OilCheck, and OilBooster, that were developed in the last year, adding the tools to the groupś already large arsenal of custom malware, ESET researchers revealed in a blog post published Dec. 14. Unique to the way the downloaders work *vs* other [PLACEHOLDER] tools is that they use various legitimate cloud services, including Microsoft OneDrive, Microsoft Graph OneDrive

API, Microsoft Graph Outlook API, and Microsoft Office EWS API, for command-and-control communications (C2) and data exfiltration, the researchers said. Attack targets so far have included a healthcare organization, a manufacturing company, a local governmental organization, and several other unidentified organizations, all in Israel and most of them previous targets for the APT. The downloaders themselves are not particularly sophisticated, noted ESET researcher Zuzana Hromcov, who analyzed the malware along with ESET researcher Adam Burgher. However, there are other reasons that the group is evolving into a formidable adversary for targeted organizations, she said. The continuous development and testing of new variants, experimentation with various cloud services and different programming languages, and the dedication to re-compromise the same targets over and over again, make [PLACEHOLDER] a group to watch out for, Hromcov said in a press statement." (Truncated for space).

- **Retrieved contexts:**

  1. Falcone, R. and Lee, B. (2016, May 26). The OilRig Campaign: Attacks on Saudi Arabian Organizations Deliver Helminth Backdoor. Retrieved May 3, 2017. Sardiwal, M, et al. (2017, December 7). New Targeted Attack in the Middle East by APT34, a Suspected Iranian Threat Group, Using CVE-2017-11882 Exploit. Retrieved December 20, 2017. Threat Hunter Team. (2023, April 20). Daggerfly: APT Actor Targets Telecoms Company in Africa. Retrieved July 25, 2024.

  2. Sardiwal, M, et al. (2017, December 7). New Targeted Attack in the Middle East by APT34, a Suspected Iranian Threat Group, Using CVE-2017-11882 Exploit. Retrieved December 20, 2017. Sherstobitoff, R., Malhotra, A. (2018, April 24). Analyzing Operation GhostSecret: Attack Seeks to Steal Data Worldwide. Retrieved May 16, 2018. Lee, B., Falcone, R. (2018, July 25). OilRig Targets Technology Service Provider and Government Agency with QUADAGENT. Retrieved August 9, 2018.

  3. Sardiwal, M, et al. (2017, December 7). New Targeted Attack in the Middle East by APT34, a Suspected Iranian Threat Group, Using CVE-2017-11882 Exploit. Retrieved December 20, 2017. Lee, B., Falcone, R. (2018, February 23). OopsIE! OilRig Uses ThreeDollars to Deliver New Trojan. Retrieved July 16, 2018. Crowdstrike. (2020, March 2). 2020 Global Threat Report. Retrieved December 11, 2020.

- **Ground Truth:** OilRig.
- **GPT-3.5 Response:** The incident described in the threat report can be attributed to the Iranian advanced persistent threat group **APT34. APT34** has been active since 2014 and primarily operates in the Middle East, targeting organizations in various industries. The group is known for cyber espionage activities and has been linked to multiple incidents in the past. The custom downloaders mentioned in the report, including SampleCheck5000 (SC5k v1-v3), ODAgent, OilCheck, and OilBooster, are part of APT34's arsenal of custom malware. These downloaders leverage legitimate Microsoft cloud services for command-and-control communications and data exfiltration, making them more evasive and difficult to detect. (Truncated for space).

- **GPT-4o Response:** The cyber incident described in the threat report can be attributed to the Iranian advanced persistent threat (APT) group known as \*\***APT34**\*\*. This group has been active since around 2014 and is known for targeting organizations in the Middle East, particularly in Israel, across various sectors including healthcare, manufacturing, and government. The specific malware mentioned in the report includes several custom downloaders: \*\*SampleCheck5000 (SC5k v1-v3)\*\*, \*\*ODAgent\*\*, \*\*OilCheck\*\*, and \*\*OilBooster\*\*. These tools are designed to leverage legitimate Microsoft cloud services for command-and-control (C2) communications and data exfiltration, which aligns with the tactics used by **APT34**. In terms of MITRE ATT&CK attribution, the following mappings can be made: - \*\*Threat Group\*\*: **APT34 (also known as OILRIG)**. (Truncated for space).

- **BART Response:** Context,,,, ['Falcone, R. and Lee, B. (2016, May 26). The OilRig Campaign: Attacks on Saudi Arabian Organizations Deliver Helminth Backdoor. Retrieved May 3, 2017. Sardiwal, M, et al. (2017, December 7). New Targeted Attack in the Middle East by APT34, a Suspected Iranian Threat Group, Using CVE-2017-11882 Exploit. Retrieved December 20, 2017. Threat Hunter Team. (2023, April 20). Daggerfly: APT Actor Targets Telecoms Company in Africa. Retrieved July 25, 2024., 'Sardiwal, M, et al. (2017, December 7). New Targeted Attack in the Middle East by APT34, a Suspected Iranian Threat Group, Using CVE-2017-11882 Exploit. Retrieved December 20, 2017. Sherstobitoff, R., Malhotra, A. (2018, April 24). Analyzing Operation GhostSecret: Attack Seeks to Steal Data Worldwide. Retrieved May 16, 2018. Lee, B., Falcone, R. (2018, July 25). OilRig Targets Technology Service Provider and Government Agency with QUADAGENT. Retrieved August 9, 2018. (Truncated for space).

- **Flan-T5 Response:** [PLACEHOLDER] has repeatedly targeted several Israeli organizations throughout 2022 in cyberattacks that were notable for leveraging a series of custom downloaders that use legitimate Microsoft cloud services to conduct attacker communications and exfiltrate data.

From the example and the evaluation results demonstrated above, we highlight a few points related to this example and for the entire dataset in general. First, the context precision is calculated in RAGAS based on the similarity between contexts and query, and context precision in this example and in the entire dataset is low. This does not imply that the RAGIntel's retriever is ineffective because the prompt does not mention the correct answer (right attack group). At the same time, a good context should have the correct answer that is not included in the query. This is the case for all queries in the dataset. Second, the faithfulness measures the correlation between the contexts and the generated responses. BART has the highest score because the generated response has the context itself. Third, the factual correctness is 0 for all models. However, GPT responses have the correct alias of the answer (APT34 alias of OilRig). This provides an example of the attack group aliases issue above.

Generally, attack attribution is a challenging task to automate as it requires powerful reasoning and inference strategies. This dataset and its evaluation results demonstrate this.

## CTI-ATTACK results

As shown in Table 1, GPT-4o achieves the highest factual correctness score (0.186) among all evaluated models on the CTI-ATTACK dataset, indicating that its responses are more factually accurate compared to the ground truths. GPT-4o and GPT-3.5 have low noise sensitivity (0.023 and 0.029, respectively), indicating robustness to irrelevant context. However, GPT-4o's faithfulness score is low (0.069), which implies that its answers often deviate from the retrieved contexts. In contrast, GPT-3.5 demonstrates a balanced factual correctness (0.115), faithfulness (0.118), and high response relevancy (0.713) scores. It is generally more consistent in producing query-aligned and contextually faithful answers, with slightly lower factual accuracy compared to GPT-4o. BART achieves the highest faithfulness score (0.169); however, its factual correctness (0.023) and response relevancy (0.481) are lower than GPT-3.5 and GPT-4o, indicating that while it often stays close to the retrieved documents, it struggles to provide accurate and relevant information to the query. Flan-T5 shows the weakest performance overall, with both factual correctness (0.017) and response relevancy (0.074) being very low. This indicates that it frequently produces factually incorrect and irrelevant answers. Overall, the results suggest that for the CTI-ATTACK dataset, which features relatively concise attack descriptions compared to CTI-ATE and CTI-TAA datasets, GPT-3.5 offers a strong trade-off between accuracy, relevance, and faithfulness. At the same time, GPT-4o excels in factual correctness but at the expense of context adherence.

## Comparison with standalone LLMs

For fair comparison with related work, an article must meet two criteria: (1) a RAG-based approach, and (2) the use of the same or part of the datasets and LLMs. None of the available related works satisfied these criteria. Therefore, we compare the performance of RAGIntel to standalone LLMs. This assesses the value of the retrieval phase in the overall quality of the generated responses. Table 3 summarizes the F1-score of the LLMs across all datasets. F1- scores of the standalone LLMs can be compared to the factual correctness metric results in Table 1. Note that the datasets include content from before, after, or undetermined periods relative to the models' knowledge cutoff dates. The comparison with the standalone LLM results in Table 3 reveals that RAGIntel does not consistently achieve higher factual correctness scores across the datasets. For example, GPT-3.5 attains an F1-score of 0.474 on CTI-ATE, 0.210 on CTI-TAA, and 0.339 on CTI-ATTACK, compared to RAGIntel's factual correctness score of 0.150, 0.037, and 0.115 on the same datasets. This observation highlights a critical trade-off. While RAGIntel's retrieval mechanism is designed to ground responses, thereby reducing hallucinations, its overall factual correctness depends on multiple factors, such as the precision of retrieval, the coverage of the knowledge base relative to the evaluation data, and the alignment between retrieved contexts and the answer space. Consequently, pre-trained LLMs on data that closely matches the evaluation set may achieve higher F1-scores despite having a higher risk of producing unsupported claims. This underlines the importance of evaluating both factual correctness and hallucination rates when assessing RAG systems.

**Table 3** Standalone LLM performance comparison for all datasets.

| LLM | F1-score (CTI-ATE) | F1-score (CTI-TAA) | F1-score (CTI-ATTACK) |
|---|---|---|---|
| GPT-3.5 | 0.474 | 0.210 | 0.339 |
| GPT-4o | 0.217 | 0.208 | 0.226 |
| BART | 0.144 | 0.049 | 0.095 |
| Flan T5 | 0.119 | 0.140 | 0.061 |

## CONCLUSION AND FUTURE WORK

Analyzing cyber threat intelligence data is critical for proactive defense, yet its manual analysis for attack investigation is inefficient. While LLMs present a promising solution, their limitations, such as hallucinations and outdated knowledge, hinder their reliability in cybersecurity applications. Our work introduces RAGIntel, a RAG-based LLM system that integrates MITRE ATT&CK as a knowledge base to enhance attack investigation accuracy. By combining dense and sparse retrieval with reranking and compression, our system retrieves the most relevant threat intelligence content, improving response quality. We extensively evaluated RAGIntel using 339 attack investigation queries, demonstrating its effectiveness comparable to standalone LLMs. Our results highlight the benefits of integrating structured threat intelligence with advanced retrieval strategies, reducing hallucinations and improving contextual understanding. RAGIntel is publicly available, enabling further research and adoption in cybersecurity defense strategies. Future work includes expanding the knowledge base with additional CTI sources and refining the retrieval mechanism for real-time threat analysis.

## ACKNOWLEDGEMENTS

The author would like to thank the anonymous reviewers for their valuable comments. The author would like to acknowledge the use of DeepSeek for assistance in preparing LaTeX tables, debugging code-related issues, and grammatical improvements, as well as Grammarly for additional language corrections. These tools enhanced the accuracy of the work.

### Funding

This project was funded by the Deanship of Scientific Research (DSR) at King Abdulaziz University, Jeddah, under grant no. (GPIP: 1074-612-2024). The funders had no role in study design, data collection and analysis, decision to publish, or preparation of the manuscript.

### Grant Disclosures

The following grant information was disclosed by the authors:
Deanship of Scientific Research (DSR) at King Abdulaziz University, Jeddah:
GPIP: 1074-612-2024.

## Competing Interests

The authors declare that they have no competing interests.

## Author Contributions

- Abeer Alhuzali conceived and designed the experiments, performed the experiments, analyzed the data, performed the computation work, prepared figures and/or tables, authored or reviewed drafts of the article, and approved the final draft.

## Data Availability

The RAGIntel code and datasets are available in the Supplemental Files.

The original third-party dataset (CTIBench) is also available at GitHub: https://github. com/xashru/cti-bench.

## Supplemental Information

Supplemental information for this article can be found online at http://dx.doi.org/10.7717/ peerj-cs.3371#supplemental-information.

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
