# Peer review of "LLM-powered threat intelligence: a retrieval-augmented generation approach for cyber attack investigation"

_PeerJ Computer Science, doi:10.7717/peerj-cs.3371_

## Round 0.1 · original submission · Major Revisions

· Academic Editor

Major Revisions

**Language Note:** The review process has identified that the English language must be improved. PeerJ can provide language editing services - please contact us at [email protected] for pricing (be sure to provide your manuscript number and title). Alternatively, you should make your own arrangements to improve the language quality and provide details in your response letter. – PeerJ Staff

Reviewer 1 ·

Basic reporting

Analyzing cyberattacks using Cyber Threat Intelligence (CTI) data presents significant challenges due to the data's diverse formats, intricate structure, and massive scale. To overcome these issues, the authors introduce RAGIntel, a Retrieval-Augmented Generation (RAG)-based large language model framework designed to improve analysis accuracy by retrieving and incorporating structured threat intelligence into the reasoning process.

Overall, the paper is well-written and presents a promising approach. However, I have several suggestions to improve the current version:

1) The GitHub repository containing the RAGIntel implementation appears to be inaccessible. Please ensure that the link is active and publicly available.

2) I recommend adding a dedicated "Threat Model" section that clearly outlines the attacker's objectives and knowledge assumptions. This will help readers and reviewers assess the realism and practicality of the attack scenarios.

3) While the paper focuses on using Retrieval-Augmented Generation (RAG) for cyberattack investigation, recent work such as [A] has shown that RAG systems themselves are susceptible to knowledge poisoning attacks. It would strengthen the paper to acknowledge and discuss these vulnerabilities in the context of your approach.

4) The paper would benefit from a report on the computational cost of the proposed method, such as overall running time.

[A] Traceback of Poisoning Attacks to Retrieval-Augmented Generation. In The Web Conference 2025.

Experimental design

no comment

Validity of the findings

no comment

Reviewer 2 ·

Basic reporting

The paper generally uses professional English, but there are instances of awkward phrasing and grammatical errors that could be improved for clarity.

The introduction sets the context of cyberattack investigation and the limitations of LLMs, motivating the proposed RAG approach. The related work section provides a good overview of RAG performance enhancements and applications in cybersecurity, with relevant references.

Figure 2 provides good examples of CTI types, but the text references it as if it provides a comprehensive overview of all types, which it does not.

Experimental design

The research proposes a novel RAG-based LLM system for cyberattack investigation, which falls within the scope of computer science and cybersecurity.

The methodology employs sound technical approaches (hybrid retrieval, reranking, compression) but has some limitations, such as having a small evaluation dataset (110 queries total), limited to a single benchmark source and no real-world validation.

Validity of the findings

The underlying data comes from established benchmarks (CTIbenchmark). The evaluation uses multiple LLMs and comprehensive metrics (RAGAS framework), which strengthens the findings.

The comparison of F1 scores between RAGIntel's factual correctness and standalone LLMs, shown in Table 3 vs. Tables 1 and 2, requires a more nuanced discussion. The numbers suggest that RAGIntel, while potentially reducing hallucinations, might not always achieve higher raw factual correctness scores compared to models pre-trained on similar data.

Additional comments

The paper presents a valuable contribution to the field of cybersecurity and LLMs. The RAG-based approach for threat intelligence is timely and addresses crucial limitations of standalone LLMs.

---

## Round 0.2 · accepted · Accept

· Academic Editor

Accept

All concerns raised by the reviewers have been satisfactorily addressed, and I am pleased to inform you that your work has now been accepted for publication in PeerJ Computer Science.

Please be advised that you cannot add or remove authors or references post-acceptance, regardless of the reviewers' request(s).

Thank you for submitting your work to this journal. I look forward to your continued contributions on behalf of the Editors of PeerJ Computer Science.

With kind regards,

Reviewer 2 ·

Basic reporting

N/A

Experimental design

N/A

Validity of the findings

N/A